# Therapeutic Management of Bronchiectasis in Children and Adolescents: A Concise Narrative Review

**DOI:** 10.3390/jcm13164757

**Published:** 2024-08-13

**Authors:** Paola Faverio, Giovanni Franco, Valentina Landoni, Marta Nadalin, Davide Negri, Alessandro Tagliabue, Federica Acone, Francesca Cattaneo, Filippo Cipolla, Chiara Vimercati, Stefano Aliberti, Andrea Biondi, Fabrizio Luppi

**Affiliations:** 1School of Medicine and Surgery, University of Milano-Bicocca, 20854 Monza, Italy; g.franco@campus.unimib.it (G.F.); v.landoni4@campus.unimib.it (V.L.); m.nadalin@campus.unimib.it (M.N.); d.negri8@campus.unimib.it (D.N.); a.tagliabue@campus.unimib.it (A.T.); f.acone@campus.unimib.it (F.A.); f.cattaneo34@campus.unimib.it (F.C.); f.cipolla3@campus.unimib.it (F.C.); chiara.vimercati@irccs-sangerardo.it (C.V.); abiondi.unimib@gmail.com (A.B.); fabrizio.luppi@unimib.it (F.L.); 2Respiratory Unit, Fondazione IRCCS San Gerardo dei Tintori, 20900 Monza, Italy; 3Pediatrics, Fondazione IRCCS San Gerardo dei Tintori, 20900 Monza, Italy; 4Department of Biomedical Sciences, Humanitas University, 20072 Pieve Emanuele, Italy; stefano.aliberti@hunimed.eu; 5Respiratory Unit, IRCCS Humanitas Research Hospital, 20089 Rozzano, Italy

**Keywords:** bronchiectasis, children, adolescent, antibiotic, vaccine, immunization, physiotherapy

## Abstract

**Introduction**: Bronchiectasis, characterized by airway dilation, mucus hypersecretion, and recurrent exacerbations, is increasingly recognized in children and adolescents. Recent guidelines from the European Respiratory Society (ERS) and Thoracic Society of Australia and New Zealand (TSANZ) emphasize early diagnosis and optimized management. This review explores therapeutic strategies for pediatric bronchiectasis. **Materials and methods**: Our review involved a comprehensive search of English-language literature in the PubMed and EMBASE databases until December 2023, focusing on observational studies, interventions, reviews, and guidelines in pediatric bronchiectasis. **Results**: Management strategies encompass airway clearance techniques, mucoactive agents, pulmonary rehabilitation, bronchodilators and inhaled corticosteroids tailored to individual needs and age-appropriate techniques. Antibiotics play key roles in preventing exacerbations, eradicating pathogens, and managing acute exacerbations, which are guided by culture sensitivities and symptoms. Long-term antibiotic prophylaxis, particularly macrolides, aims to reduce exacerbations, although concerns about antibiotic resistance persist. Vaccinations, including pneumococcal and influenza vaccines, are crucial for preventing infections and complications. Surgery and lung transplantation are reserved to severe, refractory cases after failure of medical therapies. **Conclusions**: The optimal management of pediatric bronchiectasis requires a multidisciplinary approach, including physiotherapy, pharmacotherapy, and vaccinations, tailored to individual needs and guided by evidence-based guidelines. Further research is needed to refine diagnostic and therapeutic strategies and improve outcomes for affected children and adolescents.

## 1. Introduction

Bronchiectasis is a chronic respiratory disease characterized by persistent airway dilation, mucus hypersecretion and recurrent exacerbations. Although still underdiagnosed, particularly in children and adolescents, bronchiectasis has received increasing attention in recent years with studies showing high incidence and prevalence and relevant healthcare costs. Two international guidelines have recently been published by the European Respiratory Society (ERS) [1] in 2021 and from the Thoracic Society of Australia and New Zealand (TSANZ) [2] in 2023, and the main recommendations of the two guidelines are summarized in Table 1. In light of this, the optimization of disease management, promoting early diagnosis, appropriate etiological work-up and therapeutic approach becomes of paramount importance.

The purpose of this review is to report the therapeutic management in children and adolescents with bronchiectasis, as shown in Figure 1.

## 2. Materials and Methods

A search of relevant medical literature in the English language was conducted in Medline/PubMed and EMBASE databases including observational, interventional studies, reviews and guidelines on both children and adolescents through December 2023. However, when the clinical evidence is derived from studies conducted on adult patients, these studies were also cited. In the literature research, the authors screened more than 100 manuscripts. Keywords used to perform the research are reported in Table 2. Editorials, conference abstracts and pre-print publications were excluded.

For the purpose of this review, we will only treat cystic fibrosis (CF) in the diagnostic process of bronchiectasis, but we will not focus on the management of this disease.

All studies regarding non-cystic fibrosis bronchiectasis were included, while studies reporting only on other chronic respiratory diseases whose pathogenesis is not directly related to bronchiectasis, such as asthma and bronchiolitis, were excluded. Relevant abstracts and articles were searched and screened independently by 4 authors (PF, GF, AT and CV), and when there was a discrepancy between the authors, the articles were collectively discussed analyzing relevance, strengths, and limitations.

## 3. Airway Clearance Techniques, Mucoactive Agents and Pulmonary Rehabilitation Programs

Airway clearance techniques (ACT) are widely prescribed in young patients with bronchiectasis. As reported in ERS clinical practice guidelines [3], specialized physiotherapists focusing on pediatric respiratory care are essential for managing bronchiectasis in children and adolescents, and individualized therapy must be taught and reviewed at least biannually. Particularly during exacerbations, more intensive ACT would be beneficial and may require adjustment to fit the circumstances.

Another cornerstone of pediatric respiratory physiotherapy is that the choices of techniques and devices must be appropriate to the children’s age in order to optimize the efficacy and adherence. Among the main techniques, we acknowledge those that do not require specific devices (e.g., active cycle of breathing techniques—ACBT—and autogenous drainage) and those requiring external systems or devices. Every form of ACT aims to improve ventilation and clear secretions from the airways [4]. For example, autogenic drainage (AD) is a technique that involves breathing at different lung volumes. By promoting high expiratory airflow, it helps to reduce mucus adhesion and facilitates the clearance of the secretion. The speed of the expiratory flow is essential in AD to achieve optimal results. For adolescents and children able to control their breath, the ACBT should be performed as AD. The ACBT consists in thoracic expansion, with deep controlled inspiratory exercises, and the Forced Expiratory Technique (FET). The FET is the key of ACBT. During the forced expiration, secretions are mobilized until the smaller airways by controlled huffs at low, mid, and high lung volumes. For infants and toddlers, gravity-assisted drainage (GAD) may be a useful option. It works by placing the patient in specific positions which enables gravity to drain excessive secretions from different bronchopulmonary segments.

One of the most prescribed devices, particularly suitable for children less than 4 years old, is the positive expiratory pressure (PEP) mask: a one-valve device that allows unrestricted inspiratory flow and creates a resistance to expiratory flow, generating a PEP. In this way, an increased volume of air is accumulated behind secretions, which generates a pressure gradient that forces mucus plugs toward the larger airways. Oscillating Positive Expiratory Pressure (OPEP) therapy combines PEP with high-frequency oscillations, and it is also a valid alternative. Devices like the Flutter^®^ and Acapella^®^ are commonly used. These devices generate vibrations in the airway wall to shear secretions and displace them from peripheral to central airways. The pros of these devices are the low cost, the easy use and transport and self-management. They are indicated in the early stages of the disease [5].

Other techniques imply the use of more complex devices. Temporary positive expiratory pressure (TPEP) delivers a pulsed flow (about 42 Hz) contrary to the exhaled air, resulting in a very low positive pressure of about 1 cm H_2_O. The vibration generated by the pulsed flow and transmitted throughout the respiratory tract helps to detach the secretions. Furthermore, thanks to the mouthpiece, the patient carries out an active non-forced exhalation. This “open glottal exhalation”, in addition to prolonging the expiratory phase, produces an acceleration of the expiratory flow. The results of this double effect is an improvement in the clearance of secretions and air trapping [5].

In Percussive Intrapulmonary Ventilation (IPPV), while the patient breaths normally, the device provides the patient with high-frequency mini-bursts of air (50–550 cycles per minute), thus creating an internal vibration in the airways and favoring the drainage of secretions. It may be used also in case of viscous secretions [5].

Expiratory flow acceleration accelerates the expiratory flow (thanks to a Venturi system), facilitating the ascent of the secretions up to the upper airways. It may not be effective in case of very dense secretions [5].

In more severe cases with ineffective cough and extreme muscle weakness or overt respiratory failure, the support of cough assists and ventilators for non-invasive ventilation may become necessary.

On the other hand, the use of mucoactive agents in children with non-CF bronchiectasis is still controversial. ERS guidelines [1] point out how the data for hypertonic saline and mannitol are equivocal. The balance probably favors administering hypertonic saline and mannitol in some, but not all patients. RhDNase, such as dronase alpha, has been shown to be clinically beneficial in CF lung disease, but they are contraindicated in non-CF bronchiectasis because of the increased risk of exacerbations and the accelerated lung function decline [6]. To date, evidence on the use of diverse mucoactive agents, such as N-acetylcisteine, is insufficient, and further trials are necessary. Nevertheless, it should be pointed out that the effect of the different techniques and devices may depend on the underlying cause of bronchiectasis and to reach a universal consensus, without taking into account personalization, is impossible, particularly in the field of ACT.

Furthermore, children with bronchiectasis often show an inactive lifestyle and exhibit significant development delays. For this reason, well-structured rehabilitation programs are needed. A recent pilot Randomized Clinical Trial (RCT) demonstrates that a play-based therapeutic exercise program can improve fundamental movement skills in children with bronchiectasis [7]. In addition, the program had a moderately positive effect on cardiorespiratory fitness. Anyway, larger trials are warranted to prove a real benefit of quality of life. Furthermore, a recent qualitative study tried to identify facilitators and barriers to physical activity from the perspectives of children and their parents. The authors concluded that programs to increase physical activity should be fun, accessible, provide opportunities for social interaction and address barriers related to exercise tolerance, perceived competence and presence of respiratory symptoms [7].

## 4. Bronchodilators and Inhaled Corticosteroids

In children who can perform spirometry (usually older than 4 years-old), the evaluation of lung function is essential for the evaluation of diseases and to exclude other possible comorbidities [8]. Spirometry is often preserved in cases of mild bronchiectasis. However, in moderate or more severe cases, spirometry can show an obstructive lung defect. It is essential to perform the pharmacological bronchodilation test in order to exclude concomitant asthma [9]. Some children may present a mixed obstructive–restrictive pattern. Although spirometry is the more extensively used technique to monitor lung function, it requires patients’ cooperation and therefore it is not useful for the majority of preschool-aged children. Other non-invasive techniques, which are possibly easier to perform and suitable for children younger than 4 years of age, that have been explored in the last decade include the forced oscillatory technique and multiple breath washout test [10,11].

Inhaled corticosteroids (ICS), short-acting beta-2 agonists (SABA) and long-acting beta2-agonists (LABA) are widely used in pediatric age. As explained in the above paragraph, despite a significant heterogeneity, the main pathophysiological mechanism for most bronchiectasis patients is an obstructive pattern [12]. However, the pathogenesis for lung obstruction is still not understood in its details. Therefore, a reasonable question is whether asthma-type medications could have a role in treatment of bronchiectasis-related symptoms and which patients may experience the highest benefit. In this respect, the literature is currently still controversial. International guidelines for adult patients do not suggest the use of bronchodilators in most cases except in particular circumstances including airflow obstruction, dyspnea unresponsive to standard care, and the co-occurrence of bronchiectasis with asthma or COPD [13,14,15,16].

Bronchodilators use, particularly SABA, should also be considered before using inhaled mucoactive agents or inhaled antibiotics or before undertaking ACTs [1,2]. In addition, there is unanimous consensus in discontinuing bronchodilators when no clinical benefit is obtained. With respect to ICS, their use is recommended in case of a co-existing diagnosis of asthma and/or eosinophilic airway inflammation [2]. Recent studies on adult bronchiectasis have identified an eosinophilic or T2-high endotype characterized by the presence of either eosinophils blood count ≥ 300 cells·µL^−1^ or oral fractional exhaled nitric oxide (FeNO) ≥ 25 dpp. In these cases, ICS and, in more severe cases, biologic therapies—particularly those targeting IL-5 signaling—may have a pivotal role [17,18].

In conclusion, considering the pediatric and adolescents’ population, the most recent literature discourages the routine use of bronchodilators and ICS in bronchiectasis, stressing the lack of evidence and suggesting a personalized approach based on comorbidities, pulmonary functional status and endotype. Despite such indications, the use of bronchodilators in clinical practice is still widespread, even in the absence of airway obstruction, and more clinical trials are needed to clarify their effective benefit [19].

## 5. Antibiotics to Prevent Pulmonary Exacerbations

One of the main objectives when managing children/adolescents with bronchiectasis is to reduce the rate of exacerbations and of consequent complications while aiming to improving quality of life. The use of macrolides, in a long-term low-dose schedule, is recommended by the ERS guidelines in order to minimize exacerbations [1]. Notably, macrolides also display significant immunomodulatory properties in several lung diseases, from severe asthma to bronchiolitis obliterans [20,21]. The only available multicenter, double-blind, RCT demonstrated that once-weekly azithromycin (30 mg/kg) for up to 24 months reduced pulmonary exacerbations in indigenous Australian children with bronchiectasis unrelated to CF and at least one pulmonary exacerbation in the previous 12 months [22]. Nevertheless, the development of azithromycin resistance was significantly higher in children in the azithromycin group compared to placebo. The antibiotic resistance was particularly evident for *S. aureus*, in which the resistant strains did not reduce after 6 months from the antibiotic administration, while for *S. pneumoniae*, the antibiotic resistance declined from 79% to 7% at the same time frame [23]. However, the problem of the antibiotic resistance was partially overcome by a strict adherence and compliance with the prophylaxis schedule [23]. Similarly, a recent meta-analysis concluded that therapy with macrolides prevents the exacerbation of bronchiectasis in children but with an increased risk of antibiotic resistance. The authors suggested that benefits and risks should be weighed on a patient-by-patient basis.

As for the pediatric and adolescent population, the ERS recommends long-term antibiotics for patients who undergo more than one hospitalized, or more than two non-hospitalized, exacerbations in the previous 12 months, despite the optimization of bronchiectasis management. The duration of prophylaxis should be at least 6 months, but the risk–benefit ratio for continuation needs to be periodically reassessed. The pre-treatment execution of an electrocardiogram is necessary only in the case of positive family history of prolonged-QT syndrome, arrhythmias and acute cardiac events. When possible, the research of nontuberculous mycobacteria (NTM) on lower airway specimen should be performed to exclude their presence before starting the therapy. Further studies on optimal dose and regimen are needed.

## 6. Antibiotics for Pathogens Eradication and Treatment of Exacerbations

An exacerbation is defined by an increase in cough with or without changes in the quantity or quality of the sputum associated with dyspnea, wheezing, chest pain or hemoptysis; there may also be systemic symptoms such as fever, fatigue or loss of appetite. Even though exacerbations can be induced both by viral or bacterial pathogens, all pediatrics guidelines recommend the use of antibiotics. The most common isolated bacteria are *H. influenzae*, *S. pneumoniae* and *M. catarrhalis*, while *P. aeruginosa* is more common in children with comorbidities or advanced lung damage. The choice of the appropriate antibiotic should always be based on previous isolations on sputum culture, if available, and local epidemiology. If there is no evidence or suspect of *P. aeruginosa* infection, a 14-day treatment with amoxicillin–clavulanate is the first-line treatment followed by azithromycin in case of resistance or allergy [24,25]. In case of chronic *P. aeruginosa* infection, first-line therapy implies ciprofloxacin or aminoglycosides or colistin based on culture sensitivities [1]. Intravenous antibiotics are reserved for severe exacerbations or when symptoms/signs persist more than four weeks after the use of oral antibiotics.

When *P. aeruginosa* is isolated for the first time, a trial of eradication is recommended (with very low-quality evidence for lack of direct studies), since the colonization is associated with a major risk of exacerbations, hospitalization, and disease progression [26,27].

The management approach recommended by ERS [1] and TSANZ [2] guidelines when *Pseudomonas aeruginosa* is first or newly isolated in a child with bronchiectasis depends on the specimen type and the presence of symptoms [1,2]. If symptomatic, ERS guidelines suggest initiating two weeks of intravenous antibiotics (piperacillin–tazobactam or ceftazidime + tobramycin) followed by 4–12 weeks of inhaled antibiotics such as colistin or tobramycin [1,2]. Subsequent lower airway specimen testing is recommended with consideration for repeating steps if *P. aeruginosa* persists [1,2]. If asymptomatic, guidelines recommend a two-week course of oral ciprofloxacin and/or inhaled antibiotics for 2 weeks followed by 4–12 weeks of inhaled antibiotics [1,2]. Again, lower airway specimen testing is advised, and if *P. aeruginosa* persists or symptoms develop, intravenous antibiotics followed by inhaled antibiotics are recommended [1,2].

There is a paucity of evidence on the optimal duration of antibiotics both during exacerbations and chronic maintenance therapy. Evidence is mainly derived from adults and based on expert consensus, and recent guidelines recommend antibiotic durations of approximately 14 days in acute exacerbations [28].

## 7. Vaccines

Bronchiectasis can develop following severe lower airway infections, while acute exacerbations are a predictor of progression and severity of the disease. For these reasons, preventing infection is key to managing bronchiectasis and preventing secondary complications [29]. Multiple interventions have been proposed by the different guidelines to enhance the immune defenses of children and include both the promotion of breastfeeding and full immunization according to the national programmer, including pneumococcal, pertussis and annual seasonal influenza vaccines [1,2].

*S. pneumoniae* is often present in the lower airways of children with bronchiectasis. These patients may present a deficiency in antibody response to *S. pneumoniae*, which may cause a more severe clinical phenotype and uncontrolled disease [30]. The anti-pneumococcal vaccine has been introduced in 2001, reducing the incidence of disease both in adults and children. Currently, there are two classes of vaccines. The first one consists of the 23-valent pneumococcal polysaccharide vaccine (PPV-23), which contains the purified pneumococcal polysaccharide capsular antigens of 23 serotypes. While one RCT showed that PPV-23 vaccination significantly reduced the number of acute respiratory exacerbations in adults with COPD, there are no RCTs targeting children with bronchiectasis. The second class of vaccines, the pneumococcal conjugate vaccine (PCV), contains 7, 10, 13, 15 or 20 capsular polysaccharide antigens conjugate to a carrier protein. Whilst the impact on invasive pneumococcal disease and pneumonia in children and adults has been demonstrated, the evidence for the efficacy of PCV in children and adults with bronchiectasis is lacking [31,32].

*H. influenzae* is consistently the most common bacterial pathogen isolated from the lower airways of children with bronchiectasis, although the majority *H. influenzae* are likely to be non-typeable (NTHi) strains for which there is no vaccine. Currently, the European Center for Disease Control and prevention (ECDC) and the US CDC recommend that all infants and toddlers should receive three or four doses, depending on the brand, of the *H. influenzae type B* vaccine (approximately at 2 months, 4 months, 6 months and 12 to 15 months of age). The purpose of this polysaccharide conjugate vaccine is to protect from invasive *H. influenzae type B*-related diseases, such as meningitis and pneumonia. No specific recommendations are available for children older than 5 years of age and adolescents with bronchiectasis. However, given the above-mentioned recommendations, the great majority of children in Europe and the US should be immunized against *H. influenzae type B*.

Influenza viruses are associated with acute exacerbations of chronic lung diseases in children and adults. The results of studies on the safety and effectiveness of influenza vaccines in healthy children, including those under the age of 2, are generally positive and demonstrate the effectiveness of these vaccines in reducing both acute respiratory infections and hospitalizations due to pneumonia in children [33,34,35,36,37].

Respiratory syncytial virus (RSV) is a highly contagious virus that, in children, particularly those with chronic underlying conditions such as bronchiectasis, can cause mild to severe respiratory diseases and may lead to hospitalization, ICU admission and death due to incomplete natural immunity [38]. The main characteristics of RSV infection in infants are wheezing and cough that may worsen with the development of respiratory failure. In the past, supportive care has been the only available approach until the development of specific recombinant monoclonal antibodies, Palivizumab and Nirsevimab, which have been used in recent years to protect these infants. Immature immunity and maternal antibodies protection lead to difficulties in the development of an RSV vaccine for infants. Young infants may also be protected by passively transferred antibodies in immunized pregnant women. Vector, subunit and nucleic acid approaches has been avoided in children without previous infection of RSV [39]. Palivizumab is indicated for the prevention of serious lower respiratory tract disease requiring hospitalization caused by RSV in children at high risk for RSV disease. It is usually given once a month during anticipated periods of RSV infection risk in the community (e.g., from October to March or from November to April) [40]. In 2023, Nirsevimab has also been approved for use in term and preterm babies in the EU [41]. Initially, it was approved only for adults but later showed high effectiveness in preventing low-respiratory tract infections, hospitalizations, and ICU admission also in children. It can be used also in young infants (<6 months) during their first RSV season and children between 8 and 19 months old who are at increased risk for severe RSV disease and entering their second RSV season [42]. Nirsevimab is given as a single injection before the RSV season starts or at birth for infants born during the RSV season.

Live-attenuated vaccines for the active immunization of older infants (>6 months) have been approved in more recent years. A small, 2024, double-blind, placebo-controlled trial on RSV-seronegative children showed that nasal vaccine RSV/6120/ΔNS2/1030s was immunogenic and genetically stable with increased rhinorrhea of mild entity as the only demonstrated side effect [43].

## 8. Surgery and Transplant

Surgery is uncommon and should only be considered after maximal medical therapies have failed. The evidence of benefits is still not fully demonstrated since surgery has been conducted only in patients with localized bronchiectasis but severe disease. The best results of lobectomy were registered in specialized centers. Lung transplant is rarely needed in pediatric bronchiectasis if compared with CF children. Age, severity of the disease and bronchiectasis localization should always be considered when referring the patient to surgery, since benefits are higher in patients where complete resection can be performed and when disease is not recurrent [44,45,46]. For this reason, the careful exclusion of systemic causes of bronchiectasis, including, for example, alpha1 anti-trypsin deficiency and ciliary dyskinesia, and treatment of predisposing factors or comorbidities must be performed to avoid the recurrence of bronchiectasis after surgery. Multiple indicators for referral to surgery for children with bronchiectasis have been identified and include (i) poor symptoms control despite optimization of patients’ management, (ii) poor development, and (iii) severe or recurrent hemoptysis not controlled by bronchial artery embolization [47]. Controindications for surgical referral should also be acknowledged: (i) minimally symptomatic disease, (ii) widespread bronchiectasis, (iii) ongoing acute lower respiratory tract infection, (iv) children < 6 years of age, and (v) progressive or uncontrolled underlying disease causing bronchiectasis [48].

## 9. Conclusions

The optimal management of pediatric bronchiectasis requires a multidisciplinary approach and should include a pediatrician with expertise in the pulmonary and infectious diseases field and a respiratory physiotherapist. Multiple open issues still need to be solved: first of all, there are few RCTs in children, and most of the evidence is imported from adults. Secondly, late diagnosis favors the inappropriate use of antibiotics and onset of antibiotic resistance. Thirdly, for younger children, some treatments, such as respiratory physiotherapy, may not be feasible. All these and other open issues need to be addressed by future studies.

## Figures and Tables

**Figure 1 jcm-13-04757-f001:**
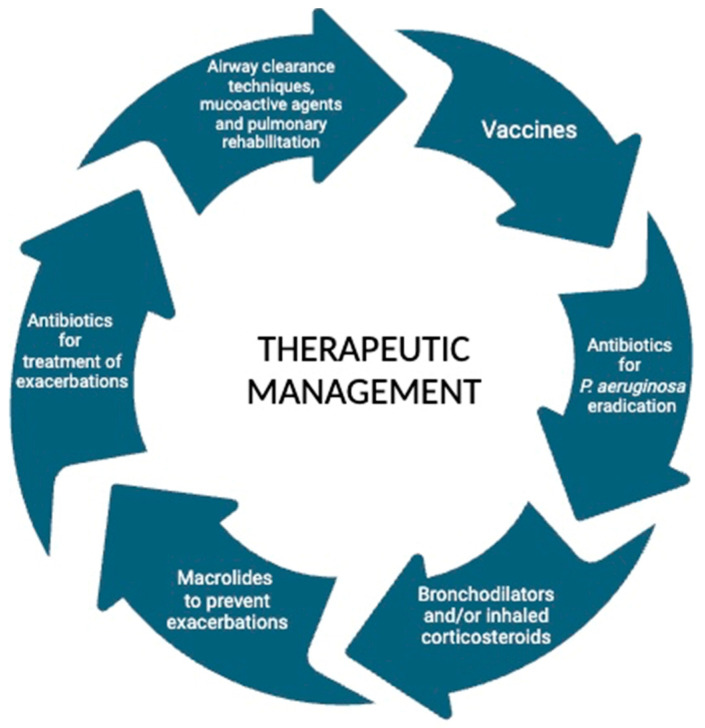
Main therapeutic interventions for children and adolescents with bronchiectasis.

**Table 1 jcm-13-04757-t001:** Main recommendations for the therapeutic management from the European Respiratory Society 2021 and the Thoracic Society of Australia and New Zealand 2023 guidelines.

ERS 2021	TSANZ 2023
**Airway clearance techniques and pulmonary rehabilitation**	
The guidelines recommend that children/adolescents with bronchiectasis are taught and receive regular individualized ACT or maneuvers (strong recommendation, low quality of evidence). The guidelines suggest that exercise is encouraged on an ongoing basis (conditional recommendation, very low quality of evidence stemming from the narrative review). There is insufficient evidence to make a recommendation for establishing formal exercise and rehabilitation programs.	Individualized ACTs are recommended. Regular physical activity is recommended for children/ adolescents with bronchiectasis.
**Mucoactive agents**	
The guidelines recommend that recombinant human DNase (rhDNase) is not used routinely (strong recommendation, very low quality of evidence). The guidelines suggest that neither bromhexine nor inhaled mannitol nor hypertonic saline are used routinely (conditional recommendation, very low quality of evidence).	Mucoactive agents, including inhaled isotonic saline, hypertonic saline and mannitol, are currently not recommended routinely.
**Bronchodilators and inhaled corticosteroids**	
The guidelines suggest not using ICS with or without LABA routinely in either the short- or long-term, irrespective of stability or exacerbation (conditional recommendation, very low quality of evidence). The guidelines underline that ICS may be beneficial in those with eosinophilic airway inflammation and that SABA maybe beneficial as pre-airway clearance therapies.	ICS and OCS should not be prescribed routinely in either the short or long-term unless there is an established diagnosis of co-existing asthma and/or eosinophilic airway inflammation. Inhaled bronchodilators should not be prescribed routinely and instead used only on an individual basis (e.g., before undertaking ACTs).
**Antibiotics to prevent pulmonary exacerbations**	
The guidelines recommend treatment with long-term macrolide antibiotics to reduce exacerbations in children and adolescents with bronchiectasis and recurrent exacerbations, particularly in those who have had more than one hospitalized or three or more non-hospitalized exacerbations in the previous 12 months (strong recommendation, low quality of evidence.)	Long-term oral macrolides should be considered for patients with the frequent exacerbator phenotype (≥3 exacerbations requiring antibiotics in the preceding 12-months) and without contraindications to these agents.
** Antibiotics for pathogens eradication and treatment of exacerbations **	
The guidelines suggest eradication therapy following an initial or new detection of *P. aeruginosa* (conditional recommendation for the intervention, very low quality of evidence). The guidelines recommend a systemic course of an appropriate antibiotic is used for 14 days in children/adolescents with bronchiectasis and an acute respiratory exacerbation. (strong recommendation, moderate quality of evidence). The guidelines underline that when the exacerbation is severe (e.g., presence of hypoxia) and/or when the child/adolescent does not respond to oral antibiotics, intravenous antibiotics will be needed.	When *P. aeruginosa* is newly detected, eradication therapy should be offered. In patients not requiring parenteral antibiotics for an acute exacerbation, oral antibiotics are prescribed for at least 14 days based upon available airway microbiology results. Close follow-up to assess treatment response is necessary. Patients failing oral antibiotic therapy for an acute exacerbation should receive intensive airway clearance strategies and parenteral antibiotics based upon the latest lower airway culture results. In children/adolescents, this requires supervised treatment for at least 14 days.
**Vaccines**	
The guidelines suggest that children/adolescents with bronchiectasis are fully immunized according to their national immunization programs, including pneumococcal and annual seasonal influenza vaccines if these are not part of this program (conditional recommendation, very low quality of evidence stemming from the narrative review).	Vaccinate according to National Immunization Program Schedules, which include pneumococcal and influenza vaccine recommendations for high-risk patients with chronic lung disorders.
** Surgery and transplant **	
Surgery is only considered after maximal medical therapies (e.g., ACT, long-term antibiotics, etc.) have failed and the child/adolescent’s quality of life remains significantly impaired. When contemplated, a multidisciplinary approach is essential, and the decision should be based on the individual’s clinical state and local surgical expertise. (Strong recommendation, very low quality of evidence stemming from the narrative review).	Although surgery is not indicated normally, there may be circumstances requiring assessment by a multidisciplinary team expert in bronchiectasis.

ACT: airway clearance techniques; ICS: inhaled corticosteroids; LABA: long-acting beta agonists; OCS: oral corticosteroids.

**Table 2 jcm-13-04757-t002:** Keywords used to perform the research.

Bronchiectasis AND airway clearance techniques AND (children AND adolescents), Bronchiectasis AND mucoactive agents AND (children AND adolescents), Bronchiectasis AND pulmonary rehabilitation AND (children AND adolescents), Bronchiectasis AND bronchodilators AND (children AND adolescents), Bronchiectasis AND inhaled corticosteroids AND (children AND adolescents), Bronchiectasis AND inhaled therapy AND (children AND adolescents), Bronchiectasis AND treatment of pulmonary exacerbations AND (children AND adolescents), Bronchiectasis AND pathogens eradication AND (children AND adolescents), Bronchiectasis AND macrolides AND (children AND adolescents), Bronchiectasis AND preventions of exacerbation AND (children AND adolescents), Bronchiectasis AND (*S. pneumoniae* OR *H. influenzae* OR *influenza virus* OR *respiratory syncytial virus*) vaccines AND (children AND adolescents), Bronchiectasis AND thoracic surgery AND (children AND adolescents), Bronchiectasis AND transplant AND (children AND adolescents)

## Data Availability

All data are available in the manuscript. No dataset have been used to write this manuscript.

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
