# Peer review of "Therapeutic Management of Bronchiectasis in Children and Adolescents: A Concise Narrative Review"

_jcm, 2024, doi:10.3390/jcm13164757_

Round 1

Reviewer 1 Report

Comments and Suggestions for Authors

Although this manuscript does not add anything new to the existing literature, it is well-written and summarizes the international guidelines for the treatment of bronchiectasis in children and adolescents. It therefore deserves our attention. 

Minor comments:

1.     On page 5, lines 224-233: H. influenzae is indeed the most common bacterial pathogen isolated from the lower airways of children with bronchiectasis. However, most H. influenzae are likely to be non-typeable (NTHi) strains for which there is no vaccine. I believe the authors should clarify this in the manuscript.

2.     On page 5, lines 250-251: Nirsevimab is not the first recombinant monoclonal antibody used for immunoprophylaxis of premature and other high-risk infants from RSV infection. Palivizumab (Synagis), another recombinant monoclonal antibody, has been used for years to protect these infants. Its main disadvantage is that it requires monthly administration from October to March (or from November to April), a period during which there is an increase in RSV infections. Please rephrase accordingly. Also, there is a spelling mistake in Nirsevimab in line 251.

3.     On page 6, line 275: Correct “need sto be solved” to “needs to be solved” 

Reviewer 2 Report

Comments and Suggestions for Authors

Dear Author/s
It has been a pleasure to read your current paper.
The manuscript is very well structured and includes usefull information on
management of bronchiectasis in children and adolescents

Your review is well documented and addresses a useful topic in paediatric pneumology, mainly cystic fibrosis and non-cystic fibrosis bronchiectasis management.

I appreciated the concise structure of the paper, which included all relevant aspects on the treatment of bronchiectasis in children and adolescents.

In the introduction you presented a very good comparison between the two current available guidelines on this topic: the recommendations for the therapeutic management from the European Respiratory Society from 2021 and the Thoracic Society of Australia and New Zealand 2023 guidelines. Table 1 is both concise and complete, offering a synthetic perspective of the similarities and differences between the 2 guidelines. Figure 1 is clear, easy to follow and evocative, but it does not include mucoactive agents and surgical procedures. These last two aspects are included in the review, and are also included in table 1.

The methods used for the search of relevant medical literature is very good described, with many details. The only observation would be that you did not provide any details on the number of papers that are reviewed.

There are some aspects that could be improved, and maybe some more references on the same topic to be included. For example, in a recent review on airway clearance techniques there are several other methods that are described, and that are used extensivelly in general practice, which the authors did not mentioned in this manuscript. (Belli S, Prince I, Savio G, Paracchini E, Cattaneo D, Bianchi M, Masocco F, Bellanti MT and Balbi B (2021) Airway Clearance Techniques: The Right Choice for the Right Patient. Front. Med. 8:544826. doi: 10.3389/fmed.2021.544826)

Regarding the use of bronchodilatators is somewhat contradictory, although some

patients with bronchiectasis many show airflow obstruction, air trapping and restriction (Cazzola M, Martínez-García MÁ, Matera MG. Bronchodilators in bronchiectasis: there is light but it is still too dim. Eur Respir J 2022; 59: 2103127)

Complete pulmonary function testing is recommended mainly in adults, but there are recent data on the benefit of spirometry in children with bronchiectasis as shown in a recent papaer (Ramsey KA and Schultz A (2022) Monitoring disease progression in childhood bronchiectasis. Front. Pediatr. 10:1010016). Some othre authors report the role of spirometry in these children (Goyal V, 2022) and others proved bronchial hyperreactivity (Guran T 2008)

The use of antibiotics and vaccination are are discussed with sufficient data regarding the indication and effectiveness, less regarding the duration and particular circumstances (for example the attitude towards the previously unvaccinated patient)

Lines 268-279 containe information on surgery without many arguments in favor or situations in which this approach is highly recommended.

I appreciated the concise review in a well structured manner of the current data on management of bronchiectasis in children and adolescents.

I think it would be necessary to add other bibliographic references, in order to be as complete as possible.

If accepted for publication, it may contribute to a better systematization of current knowledge on bronchiectasis and a most efficient method of selecting different candidate patients for speciffic treatment.

With all best wishes
